# Exploring an Intervention to Enhance Positive Mental Health in People with First-Episode Psychosis: A Qualitative Study from the Perspective of Mental Health Professionals

**DOI:** 10.3390/healthcare13151834

**Published:** 2025-07-28

**Authors:** Júlia Rolduà-Ros, Antonio Rafael Moreno-Poyato, Joana Catarina Ferreira Coelho, Catarina Nogueira, Carlos Alberto Cruz Sequeira, Sónia Teixeira, Judith Usall, Maria Teresa Lluch-Canut

**Affiliations:** 1Institut de Recerca Sant Joan de Déu, Santa Rosa 39-57, 08950 Esplugues de Llobregat, Spain; julia.roldua@sjd.es (J.R.-R.); judit.usall@sjd.es (J.U.); 2Parc Sanitari Sant Joan de Déu, Antoni Pujades 42, 08830 Sant Boi de Llobregat, Spain; 3Department of Public Health, Mental Health and Maternal and Child Health Nursing, Faculty of Nursing, University of Barcelona, 08907 L’Hospitalet de Llobregat, Spain; tlluch@ub.edu; 4Research Group on Mental Health, Psychosocial and Complex Care (NURSEARCH-2021 SGR 1083), Department of Public Health, Mental Health and Maternal and Child Health Nursing, Faculty of Nursing, University of Barcelona, 08907 L’Hospitalet de Llobregat, Spain; 5Faculty of Health Sciences, Northern School of Health of the Portuguese Red Cross, Rua Da Cruz Vermelha, 3720-128 Oliveira de Azeméis, Portugal; joana.coelho@essnortecvp.pt (J.C.F.C.); catarina.nogueira@essnortecvp.pt (C.N.); 6Research and Development Unit, Northern School of Health of the Portuguese Red Cross, Rua da Cruz Vermelha, 3720-128 Oliveira de Azeméis, Portugal; 7Center for Health Technology and Services Research—Health Research Network (CINTESIS@RISE), Rua Dr. António Bernardino de Almeida 830, 844, 856, 4200-072 Porto, Portugal; carlossequeira@esenf.pt (C.A.C.S.); soniateixeira@ufp.edu.pt (S.T.); 8Department of Biomedical Sciences, Institute of Biomedical Sciences Abel Salazar, University of Porto, Rua Jorge Viterbo Ferreira 228, 4050-313 Porto, Portugal; 9Faculty of Nursing, Nursing School of Porto, Rua Dr. António Bernardino de Almeida 830, 844, 856, 4200-072 Porto, Portugal; 10Fernando Pessoa Higher School of Health, Rua Dr. António Bernardino de Almeida 830, 844, 856, 4200-072 Porto, Portugal; 11Faculty of Health Sciences, University Fernando Pessoa, Rua Delfim Maia Street, 334, 4200-256 Porto, Portugal

**Keywords:** positive mental health, first-episode psychosis, intervention

## Abstract

**Background/Objectives:** This study explores the perspectives of mental health professionals on tailoring the Mentis Plus intervention to enhance positive mental health (PMH) in individuals experiencing First-Episode Psychosis (FEP). Although the Mentis Plus Program has been previously implemented in other contexts, it has not yet been applied to FEP care. Therefore, this study aimed to adapt the intervention for future implementation through expert consultation. **Methods:** A qualitative exploratory-descriptive design was employed. Data were collected via three focus groups comprising multidisciplinary professionals experienced in FEP care. Qualitative content analysis was used to examine the data. **Results:** Participants viewed the tailored Mentis Plus intervention as a valuable, recovery-oriented tool. Key recommendations included a flexible, group-based format with eight weekly sessions. Suggested intervention components encompassed gratitude journaling, emotional regulation techniques, and collaborative problem-solving exercises. Group delivery was highlighted as essential for mitigating isolation and promoting peer support. Practical implementation strategies included phased session structures and routine emotional check-ins. Identified barriers to implementation included the need for specialized training, limited therapeutic spaces, and the heterogeneity of participant needs. Facilitators included a person-centered approach, institutional backing, and sufficient resources. **Conclusions:** The findings support the feasibility and clinical relevance of a tailored Mentis Plus FEP Program—Brief Version. Expert-informed insights provide a foundation for adapting mental health interventions to early-psychosis care and inform future research and implementation strategies.

## 1. Introduction

First-Episode Psychosis (FEP) represents a critical and often distressing stage in the trajectory of individuals with psychotic disorders. This early phase is characterized by the onset of psychosis, including symptoms such as hallucinations, delusions, and disorganized thinking, which can profoundly affect an individual’s sense of reality and ability to function in daily life [1]. FEP also represents a high-risk period for the development of long-term psychiatric conditions, influenced by a complex interplay of biological, psychological, and social factors that shape the course of the illness. Beyond the immediate clinical challenges, individuals experiencing FEP often face significant psychological and social burdens, navigating a world that frequently stigmatizes those with mental health conditions. The stigma surrounding psychosis involves both societal rejection and personal feelings of inadequacy or shame, which further exacerbate the distress individuals face and hinder their recovery [2,3].

During this critical period, individuals with FEP frequently encounter social stigma, which can evolve into self-stigma. This internalization of societal rejection affects one’s identity and self-esteem, with profound implications for overall well-being. The experience of self-stigma can lead to feelings of hopelessness, isolation, and disengagement from treatment and social support networks [4]. As a result, individuals may be less likely to seek help, adhere to prescribed treatment regimens, or form meaningful social connections—all vital components of the recovery process [5]. This negative cycle significantly hampers recovery, diminishing quality of life, and adversely affecting long-term functional outcomes [6]. As such, early intervention is critical for mitigating these harmful effects, supporting recovery, and improving both clinical and functional trajectories for individuals with FEP [7,8].

To address these challenges, the construct of positive mental health (PMH) has been recognized in the literature as an effective framework for enhancing psychological well-being and resilience. One of the first authors to explore this concept was Jahoda [9], who proposed a set of criteria for PMH. Later, Lluch [10] developed a multifactor model that explained PMH on the basis of six interrelated factors: personal satisfaction (F1), prosocial attitude (F2), self-control (F3), autonomy (F4), resolution of problems and self-actualization (F5), and interpersonal relationship skills (F6).

Research has consistently shown that individuals experiencing FEP exhibit lower levels of these PMH factors compared with the general population [11,12]. These deficiencies hinder their ability to manage symptoms, maintain fulfilling relationships, and engage with treatment plans. Specifically, individuals with FEP often struggle with emotional regulation, impulse control, goal formulation, and social interactions [13]. These deficits, compounded by the challenges of psychosis, create significant barriers to successful recovery and reintegration into daily life [6,14]. As such, interventions that target these PMH factors are crucial for improving outcomes for this population [5].

Despite this, there is a lack of structured, recovery-oriented interventions that incorporate PMH principles into real-world clinical practice for individuals with FEP. Bridging this gap is essential to improving long-term outcomes and promoting well-being that extends beyond symptom management. This study responds to this need by exploring how a validated PMH intervention can be effectively tailored for routine care.

In this context, Mentis Plus, a mental health intervention program developed to promote PMH, emerges as a promising framework [15]. The Mentis Plus Intervention Program is based on Lluch’s PMHQ model [10] and offers a comprehensive approach that emphasizes enhancing well-being beyond the mere absence of illness. The program specifically targets the six PMH factors outlined in the model, including emotional regulation, self-esteem, interpersonal relationships, autonomy, problem-solving skills, and personal satisfaction. Over the years, Mentis Plus has been successfully adapted and implemented in various international settings and across diverse populations, including in individuals with severe mental health challenges. The program’s versatility has allowed it to be employed in different clinical contexts, demonstrating positive outcomes in enhancing mental health and well-being [15]. It has been particularly effective in improving self-esteem, emotional regulation, and social connectedness [15,16], which are crucial components of recovery in individuals with severe mental illness [6,14].

In the context of psychosis, particularly for individuals experiencing FEP, there remains a gap in interventions specifically aimed at enhancing PMH. While existing research underscores the importance of addressing PMH in individuals with psychosis, there are currently no interventions designed specifically for this population [17,18]. Although the Mentis Plus program has been shown to be effective in various clinical populations, its application to individuals with FEP has not yet been rigorously explored [19]. A systematic review [20] and a scoping review [19] on mental health interventions, both general interventions and those focused on schizophrenia, have emphasized the importance of integrating clinical and psychosocial dimensions of recovery. These reviews highlight the potential value of interventions such as Mentis Plus, which target PMH components, reinforcing the pressing need for tailored approaches in early-psychosis care.

In line with this, there is increasing recognition of the need to adapt existing interventions to meet the specific requirements of individuals with FEP. Although prior research has highlighted the distinct characteristics of this population, a gap remains in translating these insights into tailored therapeutic approaches. Adapting the Mentis Plus program specifically for individuals with FEP represents a crucial step toward bridging this gap, offering support that is both developmentally appropriate and responsive to their unique recovery trajectories.

In this context, our primary objective was to explore the perspectives of mental health professionals regarding the adaptation of the Mentis Plus intervention to enhance positive mental health in people with First-Episode Psychosis. The program had not yet been implemented in this population at the time of the study. Instead, this research represents an initial developmental phase aimed at informing a future, context-sensitive application. Specifically, this study aimed to (1) explore the concept of PMH within the context of interventions for individuals experiencing FEP; (2) analyze the perspectives of participants regarding the content and process of the tailored Mentis Plus program for this population; and (3) identify the facilitating factors and barriers encountered in the development of the Mentis Plus intervention program for individuals with FEP.

## 2. Materials and Methods

### 2.1. Design

An exploratory–descriptive qualitative study design was employed. This design is particularly suitable for the development and refinement of complex health interventions, as it enables an in-depth understanding of participants’ perspectives and contextual dynamics [21]. Data were collected through focus groups of experts in FEP, as this approach provides valuable insights into key aspects relevant to improving the well-being of individuals who have experienced their FEP. According to the literature, the use of qualitative methods such as focus groups allows researchers to explore experiences and perspectives that are crucial for constructing context-sensitive and stakeholder-informed interventions [22]. Data were collected between August 2023 and June 2024, and the study followed the Consolidated Criteria for Reporting Qualitative Research (COREQ) guidelines [23] to ensure methodological rigor and transparency. All relevant items from the 32-item COREQ checklist were addressed throughout the study’s design, data collection, and reporting phases.

### 2.2. Participants

The participants were expert professionals with extensive clinical and research experience in the field of early psychosis. Their perspectives were considered essential for identifying key factors and intervention needs for individuals experiencing FEP. Therefore, the inclusion criteria for the study called for professionals with experience in caring for individuals with FEP. Experts with specific knowledge of the diagnosis, treatment, and support of this population were selected. Participants had more than two years of experience working in mental health units or in specific FEP programs, as this period is considered sufficient to ensure meaningful clinical exposure and a solid understanding of the complexities involved in early-psychosis care [24]. While participants were not previously trained in the Mentis Plus program, some of the six professionals were familiar with its theoretical foundations and had experience applying recovery-oriented or PMH principles in clinical practice.

To select participants, professionals with expertise in FEP from the institution’s Early Psychosis Committee were invited to participate. Information about the study was disseminated through professional networks related to FEP, including expert groups and mental health professionals in both clinical practice and research. Interested individuals then directly contacted the first author, who provided further details about the study’s objectives and activities. Using purposive sampling, participants were selected based on their expertise in the field until the required number of individuals was reached to satisfy the requirements of the data collection technique while considering the predefined representativeness of participant profiles [25]. These included specialists in mental health nursing, psychologists, psychiatrists, and a social worker with expertise in FEP, both in clinical practice and research.

### 2.3. Data Collection

Data collection was carried out through three face-to-face focus groups. These expert groups played a crucial role in reviewing and adjusting the initial program content to ensure that the sessions were suitable and accessible for this population. This consistency allowed for deeper exploration of key topics and continuity across the discussions. The decision to conduct three expert focus group sessions with the same six professionals participating in each session is supported by methodological literature, which suggests that focus groups composed of 6 to 10 participants are ideal for promoting rich discussion while maintaining manageability [26,27,28]. Moreover, studies indicate that data saturation—a point at which no new information is obtained—can often be achieved with as few as three to five focus groups when participants are selected for their expertise and the topic is well defined [28,29]. Before each focus group, participants received a short summary document describing the Mentis Plus intervention (objectives, structure, and key activities) to ensure a shared baseline of understanding. This content was also reviewed briefly at the start of each session.

In our study, empirical data saturation was reached after the third focus group, as no new themes or relevant variations emerged during the final session. The participants themselves confirmed this stability, and the research team collectively agreed that the content had become redundant. This reinforced the sufficiency of the three sessions for meeting the study’s exploratory objectives.

The focus groups were facilitated by the first author, a mental health nurse specialist. The meetings lasted between 1 and 2 h and were recorded and transcribed for further analysis. A field diary (Appendix A) was used to monitor the research process both descriptively and methodologically, helping to integrate theory and practice [30,31]. At the start of each session, the group’s purpose was explained to participants, who were then provided with informed consent forms detailing how the information gathered would be used. Additionally, each group received a working document containing a specific script that outlined the topics to be discussed regarding the intervention.

The meetings followed a multi-phase feedback process. Initially, a program proposal was presented, and subsequent discussions focused on potential modifications to improve its applicability and effectiveness for individuals with FEP. This methodology took place across three key meetings: the first to present the initial proposal, the second to assess and incorporate the recommended modifications, and the third to validate the final adjustments before implementation. Once validated, the program ensured that the changes aligned with the specific needs and characteristics of individuals with FEP, with the goal of facilitating their engagement and effective participation in the program.

### 2.4. Data Analysis

For data analysis, the content analysis method [32,33] was employed. The audio of the focus groups was transcribed to capture all nuances and details from the conversations. The transcriptions were then systematically coded. In the first phase, descriptive codes were assigned based on the semantic content of the discussions. These codes were derived directly from the text, reflecting the key themes and ideas expressed by the experts.

In the second phase, the initial codes were grouped into more analytical subcategories. These subcategories classified the codes according to their meaning and the relationships between them, allowing for a deeper understanding of the participants’ perspectives on the tailoring of the Mentis Plus program for individuals with FEP.

The third phase involved a hierarchical classification process. Based on the semantic analysis of the subcategories, the codes were deductively organized according to the study’s objectives, focusing on key areas such as intervention needs, positive mental health, and effective tailoring strategies for FEP. This phase helped synthesize the experts’ input into a cohesive framework that aligned with the study’s goals.

The data analysis process followed an iterative approach, with the first and second phases being refined until a more specific understanding of the subcategories was reached. These steps were primarily carried out by the principal author. The final step, the hierarchical classification of codes, was discussed with the entire research team to ensure that the findings were robust and aligned with the research objectives.

No computer software was used in the data analysis process, as the analysis was conducted manually to maintain the depth and accuracy of the interpretation. After the third expert group discussion, the research team decided to cease further discussions, as no new significant themes emerged, indicating that data saturation had been achieved [25].

### 2.5. Rigor

Reflexivity was maintained throughout the entire process. The research team, composed of both researchers with academic training and those with practical clinical experience, enabled the establishment of a reflective and balanced approach during both the data collection and analysis. To actively minimize researcher bias, the first author kept a field diary (Appendix A) to document the research process and incorporate methodological reflections. This document was regularly reviewed and discussed with the rest of the team. Additionally, peer debriefing sessions and regular coding discussions were conducted, contributing to consistency and analytical rigor. Credibility and confirmability were strengthened through investigator triangulation during the data analysis and through informal member checking based on ongoing feedback from participants after each focus group.

### 2.6. Ethical Considerations

The study conducted by the PROFEP group, to which the present study belongs, was approved by the Research Committee of Parc Sanitari Sant Joan de Déu and the Ethics Committee of the Fundació Sant Joan de Déu (approval code PIC-148-17, approved on 1 December 2017). Accordingly, the specific study in which the tailoring of the PMH intervention to FEP is framed also received approval by the Research Committee of Parc Sanitari Sant Joan de Déu and the Ethics Committee of the Fundació Sant Joan de Déu (approval code PIC-169-23, approved on 27 October 2023), as well as by the Bioethics Committee of the University of Barcelona (approval code: CER032422, approved on 21 March 2024). Consequently, the study was conducted in accordance with the ethical standards established in the Declaration of Helsinki.

## 3. Results

The results of this study were obtained through three focus group sessions involving the same participants in each session. Specifically, six professionals took part in all three groups. The participants included one man and five women, aged between 26 and 59 years. All six participants specialized in the treatment of FEP, including two specialists in mental health nursing, two psychologists, a psychiatrist, and a social worker. They all possessed extensive experience in both clinical practice and research related to FEP.

The findings from the content analysis were grouped into three main themes: (a) the significance of an intervention to promote positive mental health in individuals with FEP, (b) the procedures for implementing this intervention, and (c) the challenges to overcome for its implementation.

### 3.1. The Significance of an Intervention to Promote Positive Mental Health in Individuals with FEP

The expert participants described the customized Mentis Plus intervention program as a highly valuable resource that, in their view, could support the emotional well-being and overall mental health of individuals experiencing FEP. They emphasized that the program appeared to help these individuals become more capable of managing emotions, strengthening their sense of self, and rebuilding social connections. Some of the six experts highlighted that, beyond symptom management, this intervention could empower individuals to focus on their personal growth and strengths. In their opinion, the approach was perceived as more positive and hopeful, especially in moments of emotional confusion and social isolation often experienced during the early stages of psychosis.
“We must help them reconnect with their identity, not only treat the illness.”—FG2P1

The experts highlighted the intervention’s unique relevance for individuals experiencing FEP, emphasizing that these individuals have specific emotional and psychosocial needs during early stages of psychosis. They emphasized that this intervention provides a critical space for individuals to safely share and process their experiences, emotions, and challenges, significantly supporting their recovery journey.
“Participants need interventions designed specifically for their experiences, allowing them to communicate openly about what they feel and what they need.”—FG2P6
“It’s vital to promote social connections and reduce self-stigma right from the beginning, helping participants develop coping strategies suitable for their situations.”—FG3P3

Additionally, the experts emphasized the significant value of group interactions in facilitating peer support, reducing feelings of isolation, and fostering a sense of community and mutual understanding among participants.
“This group setting fosters community and mutual understanding, enabling individuals to connect with others who truly understand their struggles.”—FG3P2

### 3.2. The Procedures for Implementing This Intervention

According to the experts, the recommended implementation of the adjusted Mentis Plus intervention program should be structured yet flexible and explicitly tailored for individuals experiencing their FEP. They proposed that the intervention be delivered over eight weekly sessions, each lasting approximately one hour and facilitated in a group format. This modality was considered ideal for promoting dynamic interactions, mutual support, and shared understanding among participants with similar experiences.
“Having weekly sessions in a group setting makes it easier for participants to connect and openly discuss their experiences, knowing that others share similar challenges.”—FG1P5

The group format, as highlighted by the participants, would also simplify participant connection, encourage open discussion of common challenges, reduce feelings of isolation, and enhance recruitment feasibility.

Each session, according to the experts, should focus on specific factors relevant to positive mental health within the FEP context, including personal satisfaction, prosocial behavior, self-control, autonomy, problem-solving, and interpersonal relationship skills. Suggested session activities include reflective exercises, sharing of personal strengths and challenges, emotional regulation strategies such as the STOP method (Silence, Time, Observe, Plan), and journaling tasks emphasizing achievements and gratitude.
“Activities like journaling gratitude and achievements encourage participants to reflect positively on their progress, fostering self-awareness and emotional growth.”—FG3P1
“The STOP method is simple but powerful. It gives them a way to pause and take control of their emotions before reacting.”—FG1P6

The experts confirmed that it is essential to begin each session with an open-ended question about how participants are feeling and how they have applied what was learned in the previous session.
“We should start each session by asking participants how they’re feeling and if they’ve had a chance to apply what they learned in the previous session. This would help us gauge their progress and see if the concepts are resonating with them. It’s essential that we find a way to make sure they’re reflecting on how the content is impacting them.”—FG1P1

They also suggested verifying familiarity with the PMH factor to be addressed that day, followed by a brief description to ensure clarity and engagement.
“It would be important to check if participants are familiar with the positive mental health factor we’ll focus on each day. We could give a brief description at the beginning to ensure that they know what they’ll be working on. This could help them better engage with the content and feel more prepared for the session.”—FG1P3

The session should close with a moment for personal and collective reflection on what each participant takes away from the session and how they feel about their experience.
“At the end of each session, we should ask participants to reflect on what they’ve learned from the group. This reflection could help them internalize the content and understand its relevance to their recovery. It’s crucial that we provide space for them to share how they feel about the session and what they’re taking away from it.”—FG2P1
“Having participants share their reflections in a group setting could be incredibly helpful. We could create an environment where they feel comfortable opening up, which will reinforce the learning process. Additionally, this would encourage a sense of community, which is important for their recovery journey.”—FG3P5

By addressing all factors consistently within a supportive group dynamic, the experts believed that the tailored Mentis Plus intervention program could maximize therapeutic impact, enhance participant engagement, and support sustained positive outcomes in early-psychosis recovery.

### 3.3. The Challenges to Overcome for Its Implementation

Participants identified several critical barriers that need addressing for the successful integration and implementation of the customized Mentis Plus intervention program into routine clinical practice for individuals with FEP. A primary challenge noted was the requirement for specialized training among nursing staff and facilitators. Experts stressed the necessity of developing specific competencies in positive mental health strategies, relational skills, and group facilitation techniques tailored explicitly for individuals experiencing FEP.

“We must equip professionals adequately; they need specialized training to effectively implement interventions specifically tailored for this group.”—FG3P2

Another significant barrier mentioned by experts was the limited availability of appropriate spaces and resources for group interventions. Experts highlighted the importance of creating dedicated, welcoming, and therapeutic environments to promote meaningful participation and emotional safety.

“Suitable, dedicated spaces are essential. Participants need to feel secure and comfortable in order to engage meaningfully in the intervention.”—FG2P4

Experts also recognized the complexity of accommodating diverse needs and experiences within the FEP population, presenting continuous challenges for maintaining a structured yet adaptable intervention framework.

“Balancing structure and adaptability to accommodate the diverse needs of participants requires continuous evaluation and adjustments by the facilitating team.”—FG2P3

Overall, experts concurred that addressing these challenges through targeted professional training, resource allocation, institutional backing, and continuous evaluation is essential for sustainably integrating and consolidating the customized Mentis Plus intervention program tailored specifically for individuals experiencing FEP.

To better understand the conditions required for the successful implementation of the tailored Mentis Plus intervention program in real-world clinical settings, expert groups identified both barriers and facilitators during the qualitative sessions. These insights provide practical guidance for addressing challenges related to staff readiness, infrastructure, and the adaptability of the intervention. The main findings are summarized in Table 1.

### 3.4. Final Mentis Plus FEP Program—Brief Version

Finally, the proposed Mentis Plus FEP Program—Brief Version, developed based on the contributions of the expert group, is summarized in Table 2. This program aims to address the identified needs and promote the well-being of individuals who have experienced FEP.

Finally, the modifications to the Mentis Plus FEP Program—Brief Version were agreed upon by consensus and are presented in Appendix A, which includes a session-by-session table detailing all the changes made to the original Mentis Plus intervention program. The complete tailored version of the Mentis Plus FEP—Brief Version intervention, as agreed upon by the experts, is available in Appendix A.

## 4. Discussion

This study aimed to explore an intervention to enhance PMH in individuals with FEP. Specifically, the objectives were to explore the concept of PMH within the context of interventions for individuals experiencing FEP, analyze the perspectives of participants regarding the content and process of the tailored Mentis Plus program for this population, and identify the facilitating factors and barriers encountered in the development of this intervention. In line with these objectives, the results indicated that the Mentis Plus FEP Program—Brief Version is appropriate with some modifications. These findings suggest that the intervention has the potential to enhance PMH and improve recovery outcomes for individuals experiencing FEP.

The customization of the Mentis Plus intervention program was viewed positively by the expert group, which highlighted its relevance for addressing key psychosocial challenges such as self-stigma, emotional regulation, autonomy, and problem-solving. These factors are frequently reported as being deficient in individuals experiencing early psychosis [34] and addressing them is crucial to fostering resilience and promoting recovery [6]. The tailored intervention reflected the PMH model as proposed by Jahoda [9] and Lluch Canut [10], which underscores the importance of fostering positive psychological factors such as self-control and social connectedness, elements that are particularly critical in the early stages of psychosis [35,36].

Reducing the intervention from 18 sessions to 8 weekly one-hour sessions aligns with evidence supporting the efficacy of shorter, more intensive therapeutic programs [37]. Shorter interventions have been shown to improve adherence, reduce dropout rates, and better address self-stigma in post-treatment evaluations [37]. This modification addresses the need for more accessible and feasible interventions for individuals with FEP, whose symptoms can often interfere with their ability to commit to longer treatment programs. The inclusion of specific activities, such as journaling, gratitude tasks, and structured emotional regulation techniques (e.g., STOP and the problem-solving wheel), reflects a thoughtful adequation to the symptom sensitivity commonly observed in early-psychosis populations [34]. These techniques target emotional regulation and impulse control areas that are particularly challenging for individuals with FEP [34,38,39]. Additionally, the decision to remove exercises like the mirror exercise, which could potentially trigger distressing symptoms, demonstrates a careful approach to participant vulnerability and symptom exacerbation risks.

The shift from some individual sessions to a fully group-based format was another important modification. Group interventions have long been recognized for their effectiveness in fostering social connectedness and peer support, which are crucial for reducing isolation and promoting recovery [40,41,42]. This format enables participants to engage with others who share similar experiences, normalizing their challenges and reducing feelings of self-stigma. Experts emphasized that group settings facilitate mutual understanding and support emotional resilience, key components of the recovery process. The social aspect of group therapy, particularly for individuals with FEP, is essential in normalizing experiences and building a community, which is integral to their recovery journey [43].

While these modifications are promising, the study also highlighted several challenges that need to be addressed for successful integration into clinical practice. One of the primary barriers identified by the experts was the need for substantial staff training in person-centered and recovery-oriented therapeutic approaches [5]. Given the unique needs of individuals with FEP, it is essential to equip staff with the skills necessary to implement these interventions effectively [44]. Training in positive mental health strategies and relational skills tailored to the needs of individuals with FEP is crucial for the program’s success. Additionally, the availability of suitable therapeutic spaces that foster engagement and emotional safety was identified as a key factor for successful implementation [45]. These findings underscore the importance of both professional preparation and infrastructural support in ensuring the intervention’s effective integration into clinical practice.

While the expert perspectives provided valuable insights into the relevance and adaptability of the Mentis Plus program, it is important to acknowledge that these views may differ from those of individuals with lived experience of FEP. A recent systematic review of design approaches in mental health care highlights that involving service users and people with lived experience enhances engagement, supports tailored intervention content, and fosters empowerment through participatory co-design processes [46]. Moreover, co-design research has shown that involving people with lived experience in the development and evaluation of interventions can enhance their relevance, contextual fit, and perceived usefulness [47]. Reflecting on these differences is essential when interpreting expert recommendations related to session structure, content, and delivery format.

### Strengths and Limitations of the Study

This study presents different strengths that contribute to its relevance and practical utility. Notably, the use of focus groups with experienced professionals in the treatment of FEP allowed for an in-depth exploration of implementation strategies, content refinement, and contextual needs for intervention delivery. The qualitative design enabled a rich understanding of expert perspectives, and the consistent participation of the same professionals across sessions enhanced the continuity and coherence of the insights collected. This consistency allowed for a longitudinal dialogue across sessions, enabling participants to build upon previous reflections, deepen discussions, and collaboratively validate and refine the intervention. It also reduced inter-group variability and fostered trust within the group, contributing to more open and nuanced feedback [27]. Moreover, the study is grounded in current evidence-based frameworks, such as positive mental health and person-centered care, which enhances its theoretical robustness and clinical relevance.

However, several limitations must be considered in interpreting the findings. Firstly, the intervention refinement process primarily involved expert consultations without direct input from individuals with lived experience of FEP. While expert perspectives provided valuable clinical insights and feasibility assessments, the lack of patient co-design potentially limits the ecological validity and comprehensive acceptability of the intervention. Future research incorporating direct patient engagement through qualitative and co-design methods is essential to fully capture experiential perspectives, preferences, and actual patient needs. Patient involvement could also ensure the intervention’s alignment with real-world experiences, enhancing adherence and outcomes.

Looking ahead, future studies should incorporate patient input and employ longitudinal designs to evaluate the long-term impact of this intervention on the recovery of individuals with FEP. Additionally, comparative studies with other established early intervention models could provide valuable insights into the relative effectiveness of this tailored intervention and identify areas for further refinement.

Finally, within the international context, the Mentis Plus FEP Program—Brief Version can be positioned alongside well-established early-intervention models. Drawing on the recent scoping review by Gouveia et al. [43], we relate Mentis Plus to comprehensive programs such as NAVIGATE (USA), EPPIC (Australia), and OPUS (Denmark). While these focus primarily on symptom reduction, medication adherence, and family psychoeducation, Mentis Plus adds a distinct value by explicitly promoting PMH dimensions such as autonomy, emotional regulation, and personal satisfaction. This person-centered and recovery-oriented approach may complement existing interventions by targeting psychosocial strengths early in the treatment process.

## 5. Conclusions

The Mentis Plus FEP Program—Brief Version may offer a comprehensive, person-centered approach to promoting positive mental health and supporting recovery. This tailored program represents a crucial step toward improving mental health outcomes for individuals with FEP and contributes meaningfully to the development of effective early-psychosis interventions. This version of the program demonstrates substantial promise in addressing critical dimensions of PMH among individuals experiencing FEP. Its structured yet flexible approach, tailored specifically to the vulnerabilities and strengths of this population, positions it as a potentially valuable addition to early-psychosis intervention frameworks.

## 6. Relevance for Clinical Practice

This intervention holds significant clinical implications, providing mental health professionals with a robust, evidence-informed therapeutic approach specifically tailored for early-psychosis populations. The systematic implementation of the Mentis Plus FEP Program—Brief Version could substantially contribute to enhanced coping skills, reduced self-stigma, and improved overall quality of life, ultimately supporting more favorable recovery trajectories for young adults in early-psychosis treatment contexts. For successful integration into clinical practice, ongoing institutional commitment to staff development, resource allocation, and supportive organizational policies will be indispensable. Furthermore, embedding patient feedback mechanisms into routine clinical practice could continuously inform adaptations, ensuring that the intervention remains aligned with patient needs and experiences over time.

## Figures and Tables

**Table 1 healthcare-13-01834-t001:** Barriers and facilitators for the integration of the tailored Mentis Plus intervention program into clinical practice ^1^.

Barriers	Facilitators/Recommendations	Supporting Quote
Lack of specialized training for nursing staff and facilitators.	Develop training programs focused on positive mental health, relational skills, and group facilitation tailored to FEP.	“We must equip professionals adequately; they need specialized training to effectively implement interventions specifically tailored for this group.”—FG5P2
Limited availability of suitable spaces and resources for group sessions. Diversity of needs and experiences among individuals with FEP complicates program standardization.	Create welcoming, dedicated, and therapeutic spaces to foster engagement and emotional safety. Implement continuous evaluation mechanisms to adjust the intervention framework based on participant diversity.	“Suitable, dedicated spaces are essential. Participants need to feel secure and comfortable in order to engage meaningfully in the intervention.”—FG6P4 “Balancing structure and adaptability to accommodate the diverse needs of participants requires continuous evaluation and adjustments by the facilitating team.”—FG2P9

^1^ FEP = First-Episode Psychosis; supporting quotes are drawn from focus group participants (e.g., FG5P2 = Focus Group 5, Participant 2).

**Table 2 healthcare-13-01834-t002:** Summary of the final version of the Mentis Plus FEP Program—Brief Version ^1^.

Session	Activities	PMH Factor Addressed
Initial	Presentation of participants, program, and PMH.	
	Evaluations of PMH.	
	Agree on group day and attendance. Sign IC and fill in sociodemographic variables form.	
1st Session	Explain personal satisfaction.	
	Write down a defect and a virtue, share them. Discussion about feelings. List of positive things. Share 3 gratitude statements and 3 achievements.	F1: Personal Satisfaction
2nd Session	Evaluate the activity of noting gratitude statements and achievements.	
	Explain prosocial attitudes.	
	Reflect on the song “Sentir”.	F2: Prosocial Attitude
	Express desired changes for the well-being of others. Write concrete strategies.	
3rd Session	Evaluate the first step of the first objective. Explain self-control. Write 2 recent situations of strong emotion or impulsiveness. Share situations. Explain the STOP technique. Teach relaxation technique in movement.	F3: Self-control
4th Session	Evaluate regular practice of relaxation exercises. Explain autonomy. Write strengths, opportunities, weaknesses, and threats using the SWOT technique. Explain the wheel of emotions. Write 5 achievements.	F4: Autonomy
5th Session	Explain problem-solving and self-actualization. Present a recent problem and solutions. Explain the problem-solving wheel and apply it.	F5: Problem-solving and Self-actualization
6th Session	Evaluate the experience of applying the problem-solving wheel. Explain interpersonal relationship skills. Write a difficulty, put it in a box, read and propose solutions.	F6: Interpersonal Relationship Skills
Final Session	Group reflection on the program. Evaluations of PMH and personal satisfaction. Slogan summary activity. Program closure.	

^1^ PMH = Positive mental health; IC = Informed consent; STOP = Silence, Time, Observe, Plan; SWOT = Strengths, Weaknesses, Opportunities, Threats; F = Factor.

## Data Availability

The data supporting the findings of this study are not publicly available due to confidentiality and ethical restrictions related to the privacy of the participants.

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
