# Peer review of "Exploring an Intervention to Enhance Positive Mental Health in People with First-Episode Psychosis: A Qualitative Study from the Perspective of Mental Health Professionals"

_healthcare, 2025, doi:10.3390/healthcare13151834_

Round 1

Reviewer 1 Report

Comments and Suggestions for Authors

The manuscript shows strong adherence to qualitative research standards (e.g., COREQ).

Use of purposive sampling, expert focus groups, iterative content analysis, and reflexivity is appropriate and clearly described.

However, some areas could benefit from clarification, stronger transitions, and minor language polishing.

Background

  • Why is this research necessary? The background does not answer this question. The authors should place more emphasis on the importance of intervention study in the real world.

Methods

  • The sample size for this article is small, and the sampling method is non-random sampling. How the participants were included? Do you have inclusion criteria? In short, the strength of evidence in this paper is limited.

  • Could add a bit more on how researcher bias was actively minimized beyond just triangulation.
  • Repetition of ethics approval could be tightened (lines 237–243 mention the same two committees multiple times).
  • Fix typos such as:

"social work" (line 172) → should be "social worker"

“Data was collected…” (line 148) → “Data were collected…”

  • Double-check punctuation and hyphenation (e.g., “context-sensitive,” “face-to-face”).
  • Consider adding a table summarizing participants (e.g., role, experience) to enhance transparency.
  • Include brief justification for why three focus groups were sufficient beyond literature —g., mention saturation in the data collection section, not just analysis.

Discussion

  • You mention that "the consistent participation of the same professionals" was a strength — you might briefly say why this matters (e.g., it enabled longitudinal feedback, reduced variability, and deepened trust in the group process).

Conclusion

  • Phrases like “The Mentis Plus FEP Program - Brief Version” appear twice in a short paragraph. You can vary the wording to improve readability.

Comments on the Quality of English Language

The English could be improved to more clearly express the research.

Reviewer 2 Report

Comments and Suggestions for Authors

This manuscript presents a well-conducted qualitative study aiming to tailor the Mentis Plus program for individuals experiencing First-Episode Psychosis. The authors used expert focus groups to adapt the intervention, following a rigorous and systematic methodology. The topic is highly relevant, addressing the gap in positive mental health interventions for early psychosis.

Although expert input is invaluable, direct perspectives from people with lived FEP experience are missing. Acknowledge this limitation more explicitly in the Discussion, and propose future research steps involving patient co-design.

The description of the manual content analysis could be strengthened. Add a few lines describing how inter-rater reliability or investigator triangulation was ensured beyond general mention (e.g., did multiple researchers independently code sections? Was consensus reached through discussion?)

Although the manuscript is well-referenced, it could benefit from a brief discussion of how the adapted Mentis Plus compares with other early psychosis interventions internationally. Add 1-2 paragraphs in the Discussion on similarities/differences compared to existing interventions globally.

Reviewer 3 Report

Comments and Suggestions for Authors

Overall, a very useful well written study that has clear practical application. There are a few minor comments for consideration.

The title does not seem to reflect the study. It would be beneficial if the title reflected that the study is from the perspective of healthcare professionals. Further specific information on the intervention would better reflect what the study is about. Perhaps if either the mentis plus intervention was named or more specific detail on what type of intervention is being implemented/evaluated eg Positive Mental Health.

Introduction

The introduction is well written and clearly highlights a rationale for the need of the intervention and study.

Clarity is required round the intervention and the aim of the study, for instance was this already implemented or were you looking to develop this and implement it in the future. Upon reading the paper, my understanding is that you were developing PMH for implementation – further clarity needed on this at the end of the introduction and in the aims. At times the introduction/abstract/title suggests this intervention is already implemented and you are exploring this existing intervention.

Method

COREQ – brief overview of how this was used and which, if all items on the checklist were implemented.

The participants section could be more succinct to enhance clarity. Further information on what experience of PMH/Mentis Plus the participants have (if any). If they had no experience, how was information about the intervention shared? Further information could be provided on this in the data collection section eg material shared beforehand or what was presented?

In the rigour section how was triangulation undertaken?

Results

Clear results are presented with quotes which clearly represent the overall theme highlighted.

Conclusion

Should the first line not read ‘The Mentis Plus FEP Program - Brief Version could offer a comprehensive…’ as it has not yet been implemented into practice?
